# Promising Antifungal Molecules against Mucormycosis Agents Identified from Pandemic Response Box^®^: In Vitro and In Silico Analyses

**DOI:** 10.3390/jof9020187

**Published:** 2023-01-31

**Authors:** Mariana Ingrid Dutra da Silva Xisto, Rodrigo Rollin-Pinheiro, Yuri de Castro-Almeida, Giulia Maria Pires dos Santos-Freitas, Victor Pereira Rochetti, Luana Pereira Borba-Santos, Yasmin da Silva Fontes, Antonio Ferreira-Pereira, Sonia Rozental, Eliana Barreto-Bergter

**Affiliations:** 1Laboratório de Química Biológica de Microrganismos, Departamento de Microbiologia Geral, Instituto de Microbiologia Paulo de Góes, Universidade Federal do Rio de Janeiro, Rio de Janeiro 21941-902, Brazil; 2Laboratório de Biologia Celular de Fungos, Programa de Biologia Celular e Parasitologia, Instituto de Biofísica Carlos Chagas Filho, Universidade Federal do Rio de Janeiro, Rio de Janeiro 21941-902, Brazil; 3Laboratório de Bioquímica Microbiana, Departamento de Microbiologia Geral, Instituto de Microbiologia Paulo de Góes, Universidade Federal do Rio de Janeiro, Rio de Janeiro 21941-902, Brazil

**Keywords:** *Rhizopus*, Pandemic Response Box^®^, antifungal agents, biofilm, in silico analyses

## Abstract

Mucormycosis is considered concerning invasive fungal infections due to its high mortality rates, difficult diagnosis and limited treatment approaches. Mucorales species are highly resistant to many antifungal agents and the search for alternatives is an urgent need. In the present study, a library with 400 compounds called the Pandemic Response Box^®^ was used and four compounds were identified: alexidine and three non-commercial molecules. These compounds showed anti-biofilm activity, as well as alterations in fungal morphology and cell wall and plasma membrane structure. They also induced oxidative stress and mitochondrial membrane depolarization. In silico analysis revealed promising pharmacological parameters. These results suggest that these four compounds are potent candidates to be considered in future studies for the development of new approaches to treat mucormycosis.

## 1. Introduction

Mucormycosis is an invasive and aggressive fungal infection with high mortality and morbidity, difficult diagnosis and resistance to several antifungal drugs [1]. It is caused by fungi belonging to the order Mucorales, with *Rhizopus* spp. being the most common etiologic agent. Other agents include *Mucor* spp., *Cunninghamella* spp., *Lichtemia* spp. and *Rhizomucor* spp. [2]. Immunocompromised individuals are mostly infected through the inhalation of spores and immunocompetent individuals suffering from severe burns or physical traumas can be infected by contact [3,4]. The most common clinical manifestations are rhino-orbital/cerebral and pulmonary mucormycosis. The disease is known for its necrotizing and angioinvasive nature as spores are able to adhere and form hyphae in endothelial cells; as a consequence, the host’s blood vessels are blocked, leading to the death of surrounding tissues and to hematogenous dissemination to other organs. Uncontrolled diabetes mellitus, ketoacidosis, organ transplants and neutropenia are among the several predisposing risk factors for mucormycosis [5]. During the coronavirus disease 2019 (COVID-19) pandemic, an unprecedented increase in mucormycosis cases was reported. During the months of May and August 2021, the Indian government reported over 47,500 cases of COVID-19-associated-mucormycosis (CAM) and other countries such as United States of America, Mexico, Brazil, Chile, Germany, United Kingdom and China also reported cases of CAM [6].

Mucormycosis treatment includes surgical procedures and the treatment of predisposing factors combined with systemic antifungal therapy with liposomal formulation of amphotericin B, since this drug exhibits good activity against several Mucorales species [7]. Azoles such as itraconazole, fluconazole and voriconazole usually have no effect against most mucormycosis-causing pathogens. Posaconazole and isavuconazole exhibit greater activity than other azoles, but the efficacy is species-dependent with limited activity against *Rhizopus* spp. and *Mucor* spp. [8]. Previous studies have reported that azole resistance in Mucorales is intrinsic and probably related either to conserved amino acid substitution of lanosterol 14α-demethylase, the molecular target of azoles, or to the expression of ABC transporters responsible for drug efflux [9,10]. As several Mucorales species are resistant to conventional antifungal drugs, current treatment options for mucormycosis are limited and may involve major side effects for patients. Therefore, the search for new compounds with higher efficacy against mucormycosis-causing pathogens is important.

The screening of compound libraries has emerged as a promising approach to identify new molecules with antifungal activity as well as repurposing known drugs. This proposal has been successively tested with several other fungal pathogens such as *Aspergillus* spp., *Candida* spp., *Cryptococcus* spp. And *Sporothrix* spp. [11,12,13]. In this context, the Medicines for Malaria Venture (MMV) organization provided the Pathogen Box^®^ and the Pandemic Response Box^®^ with 400 compounds each. In our previous studies, we performed the screening of the Pathogen Box^®^ and identified compounds that exhibited antifungal activity against pathogenic *Scedosporium* and *Lomentospora* species, such as auranofin and iodoquinol [14]. Considering the rise in mucormycosis cases due to the COVID-19 pandemic and the difficulty of treatment, this study aimed to screen the 400 compounds of the Pandemic Response Box^®^ in order to identify potential new compounds with antifungal activity against *Rhizopus* spp., the most common causative agent of mucormycosis.

## 2. Materials and Methods

### 2.1. Strains and Growth Conditions

*Rhizopus oryzae* UCP1295, *Rhizopus microsporus* var. *microsporus* UCP1304 and *Rhizopus stolonifer* UCP1300, isolated from the Brazilian Caatinga area, were supplied by Galba Maria de Campos-Takaki, from the Culture Collection (RENNORFUN) of the Catholic University of Pernambuco, Recife, Brazil. Fungal stocks were kept in potato dextrose medium (Neogen, Lansing, MI, USA). To obtain conidia, cells were grown on potato dextrose agar plates for seven days at room temperature. Conidia were obtained by washing the plate surface with phosphate-buffered saline (PBS, pH 7.2), and hyphal fragments and debris were removed via filtration through a Cell Strainer (Falcon, Glendale, AZ, USA). The suspension was then centrifuged and cells were counted in Neubauer’s chamber to be used in the experiments.

### 2.2. Compounds

The Pandemic Response Box^®^ library was provided by the Medicines for Malaria Venture organization and is composed of 400 compounds at 10 mM in dimethyl sulfoxide (DMSO) (Tedia, Fairfield, OH, USA). A stock solution of each compound was kept at 1 mM in DMSO and stored at −20 °C. Amphotericin B and posaconazole were used as standard antifungal drugs and were obtained from Sigma-Aldrich (Sigma Chemical Co., St. Louis, MO, USA).

### 2.3. Screening of the Pandemic Response Box^®^ Library

*Rhizopus oryzae* UCP is the most prevalent Mucorales species in human infections and was used as a reference strain to screen the Pandemic Response Box^®^ library.

Screening was performed according to Rollin-Pinheiro and colleagues [14]. Each compound was diluted in RPMI 1640 medium (Sigma Chemical Co., St. Louis, MO, USA), supplemented with 2% glucose and buffered with 3-(N-morpholino) propanesulfonic acid (MOPS) (Sigma Chemical Co., St. Louis, MO, USA) (0.165 mol/L, pH 7.2, from here on referred to as ‘supplemented RPMI’) to reach 5 μM in 96-well microtiter plates. Posaconazole (5 μM) and supplemented RPMI containing 1% DMSO were used as controls (both from Sigma Chemical Co., St. Louis, MO, USA). Conidia (2 × 10^5^/mL) were added and incubated for 48 h at 37 °C in a 5% CO_2_ atmosphere. Fungal growth was quantified via optical density readings using a spectrophotometer (Bio-Rad, Hercules, CA, USA) at 600 nm. An inhibition of at least 60% was defined as the cut-off to select the promising drugs with antifungal activity against the *Rhizopus* species.

### 2.4. Antifungal Susceptibility Testing

The susceptibility of the *Rhizopus* species to MMV396785 (alexidine), MMV1580844, MMV642550 and MMV019724 was determined using the broth microdilution method, according to EUCAST protocols, with modifications [14,15]. Posaconazole and amphotericin B were also used as reference antifungal drugs because they are commonly used for the treatment of mucormycosis. Briefly, a serial dilution (20–0.04 μM) of each compound was obtained in supplemented RPMI 1640 medium in 96-well microplates. A standardized suspension of conidia (2 × 10^5^/mL) was added to the wells and incubated for 48 h at 37 °C, in a 5% CO_2_ atmosphere. Fungal growth was analyzed via spectrophotometry readings at 600 nm and cell viability was assessed using the XTT reduction assay [16]. Minimum inhibitory concentration (MIC) of each compound was defined as the lowest concentration that inhibits 50% of fungal growth, because these fungi are highly resistant to antifungal drugs and a reduction in half of their growth is relevant. After MIC readings, MFC (minimal fungicidal concentration) was determined by subculturing, in drug-free potato dextrose agar medium, 4 µL aliquots from each well of the serial dilution, followed by spectrophotometric growth evaluation after 48 h. MFC values were defined as the lowest drug concentration able to arrest fungal growth [17].

### 2.5. Biofilm Formation and Preformed Biofilm Assay

Biofilm formation assay was performed according to Rollin-Pinheiro and colleagues [14]. Biofilms were grown on the surface of sterile polystyrene microplates (96-well). Briefly, serial dilutions of selected compounds were prepared in RPMI 1640 medium (8–¼× MIC) and deposited in 100 µL aliquots in microplate wells. Subsequently, 100 µL containing 10^5^
*Rhizopus* conidia in RPMI 1640 medium was added to each well and incubated for 48 h at 37 °C. The positive growth control did not contain any drug.

For the preformed biofilm assay, cells were cultured as described above in the absence of the compounds. After 24 h of biofilm formation, the supernatant was removed and RPMI was added either without (positive control) or with selected compounds (8–¼× MIC). An additional incubation of 24 h at 37 °C was performed to evaluate the anti-biofilm activity. Evaluation of both biofilm formation and preformed biofilms was performed using three parameters as previously described [18,19]. Crystal violet, safranin and XTT assays were used to analyze the overall biomass, extracellular matrix and metabolic activity, respectively.

### 2.6. Scanning Electron Microscopy

Scanning electron microscopy was performed according to Rollin-Pinheiro and colleagues and Borba-Santos and colleagues [14,20]. *Rhizopus oryzae* was grown in RPMI in the absence or the presence of selected compounds (½ MIC), with orbital agitation (150 rpm) for 48 h. Cells were gently collected, washed in sterile PBS and processed according to the following steps: (i). fixation in 2.5% glutaraldehyde and 4% formaldehyde, in 0.1 M cacodylate buffer, for 30 min at room temperature; (ii). washing in 0.1 M cacodylate buffer; (iii). post-fixation in 1% osmium tetroxide in 0.1 M cacodylate buffer containing 1.25% potassium ferrocyanide for 30 min; (iv). washing in 0.1 M cacodylate buffer again; (v). dehydration in a graded ethanol series (30–100%); (vi). critical point drying in CO_2_ (EM CPD300, Leica Mycrosystems, Wetzlar, Germany); (vii). adhesion to aluminum stubs with carbon tape; and viii. coating with gold. 

Images were obtained using FEI Quanta 250 (FEI Company, Hillsboro, OR, USA) and ZEISS EVO 10 (ZEISS, Oberkochen, Germany) scanning electron microscopes, and processed using Photoshop software (Adobe, San José, CA, USA).

### 2.7. Antifungal Drug Synergy Assay

Synergistic interactions were evaluated using the checkerboard method according to EUCAST guidelines (EUCAST 2008). *R. oryzae* conidia (2 × 10^5^/mL) were grown in 96-well plates containing supplemented RPMI in the presence of the following selected compounds: Alexidine (0.04–2.5 μM), MMV1580844 (0.008–0.5 μM), MMV642550 (0.008–5 μM) and MMV019724 (0.04–2.5 μM), combined with posaconazole (0.31–20 μM) or amphotericin B (0.625–40 μM). After incubation for 48 h at 37 °C, MIC was evaluated at 600 nm and cell viability was assessed via the XTT reduction assay at 490 nm. An inhibition of at least 50% was defined as the cut-off for minimum inhibitory concentration (MIC). Interactions were determined by the fractional inhibitory concentration index (FICI), which was calculated using the following formula: (MIC combined/MIC drug A alone) + (MIC combined/MIC drug B alone). The results were classified as follows: synergistic effect, FICI of ≤0.5; no effect, FICI of >0.5–4.0; antagonistic effect, FICI of >4.0 [21]. The Bliss independence model calculation was performed according to Meletiadis and colleagues and Zhao and colleagues [22,23]. The following formula was used to assess drug interaction: Eexp = Ea + Eb − Ea × Eb, in which Eexp is the expected efficacy of drug combination, Ea is the efficacy of drug A (Alexidine, MMV1580844, MMV642550 and MMV019724) and Eb is the efficacy of drug B (posaconazole or amphotericin B). The results were classified as follows: synergistic effect when Eobs > Eexp; indifference when Eobs = Eexp; antagonistic effect when Eobs < Eexp. 

### 2.8. Analysis of Fungal Cell Alterations

Alterations of *R. oryzae* cells caused by alexidine, MMV1580844, MMV642550 and MMV019724 were analyzed using fluorescent staining [20,24]. Chitin, nucleic acid, oxidative stress, mitochondrial membrane potential and neutral lipid content were evaluated using calcofluor white (Sigma-Aldrich, MO, USA), Sytox Blue (ThermoFisher, Waltham, MA, USA), 2′,7′–dichlorofluorescein diacetate (DCFH-DA) (Sigma-Aldrich, MO, USA), JC-1 probe (ThermoFisher, Waltham, MA, USA) and Nile Red (Sigma-Aldrich, MO, USA), respectively. Cells were grown in the absence (positive control) or the presence of ½ MIC of alexidine, MMV1580844, MMV642550 and MMV019724 for 48 h at 37 °C. Cells were stained with 25 μg/mL of calcofluor white, 20 μM of Sytox Blue, 50 μg/mL of DCFH-DA, 10 μg/mL of JC-1 or 20 μg/mL of Nile Red, all of them for 1 h at 37 °C in the dark. Samples were washed three times to remove residual dye and suspended in PBS. Fluorescence intensity was measured using the SpectraMax 340 microplate reader (Molecular Devices, San José, CA, USA) under the following conditions: calcofluor white at 350 nm (excitation) and 432 nm (emission); Sytox Blue at 444 nm (excitation) and 480 nm (emission); DCFH-DA at 492 nm (excitation) and 517 nm (emission); JC-1 at 475 nm (excitation) and 529 nm (green fluorescence) or 590 nm (red fluorescence) for the calculation of the red/green fluorescence intensity; and Nile Red at 550 nm (excitation) and 635 nm (emission).

### 2.9. Cytotoxicity Assay

Cytotoxicity of alexidine, MMV1580844, MMV642550 and MMV019724 was performed using three cell lineages: RAW 264.7, a murine macrophage culture; A549, adenocarcinomic human alveolar basal epithelial cells; and HaCaT, a spontaneously transformed aneuploid immortal keratinocyte cell line from adult human skin. Cells were grown in DMEM (Dulbecco’s Modified Eagle’s Medium, Sigma-Aldrich, MO, USA), supplemented with 10% fetal bovine serum and incubated for 48 h at 37 °C with 5% CO_2_ in the presence of serially diluted concentrations (0.3–50 μM) of each compound. Cell viability was measured using the neutral red (NR) assay and quantified using a spectrophotometer at 595 nm (SpectraMax^®^ i3x, Molecular Devices^®^, San José, CA, USA) [24,25].

### 2.10. In Silico Analysis

Physicochemical properties, such as molecular weight (MW), lipophilicity, as described by calculated octanol/water partition coefficient (cLogP), number of hydrogen bond donors (HBDs), number of hydrogen bond acceptors (HBAs) (set of molecular descriptors of Lipinski’s Rule of Five—RoF) and number of rotatable bonds (nRotBs), and topological polar surface area (TPSA) (both molecular descriptors of Veber’s rule) of MMV1580844, MMV642550, MMV019724 and alexidine were obtained using the open-source Molinspiration Property Calculator (Molinspiration Cheminformatics, Bratislava University, Slovak Republic; available at: https://www.molinspiration.com/cgi-bin/properties, accessed on 4 November 2022) and OSIRIS Property Explorer (Actelion Pharmaceuticals. Ltd., Allschwil, Switzerland; available at: https://www.organic-chemistry.org/prog/peo/, accessed on 4 November 2022) to predict the drug-likeness of these compounds. Since Ro5 is not applicable for natural compounds [26], amphotericin B and posaconazole were included in our analysis only for comparative purposes. The simplified molecular input-line entry system (SMILES) of the selected compounds and the standard antifungals were obtained from MMV and PubChem databases, respectively.

### 2.11. Statistical Analyses

All experiments were performed in triplicate, in three independent experimental sets. Statistical analyses were performed using GraphPad Prism v5.00 for Windows (GraphPad Software, San Diego, CA, USA). Nonparametric Kruskal–Wallis one-way analysis of variance was used to compare differences among groups, and individual comparisons of groups were performed using the Bonferroni post-test. The 90% or 95% confidence interval was determined in all experiments.

## 3. Results

### 3.1. Screening of the Pandemic Response Box^®^

The screening of the Pandemic Response Box^®^ was performed using *R. oryzae* as the standard species. The evaluation of the 400 compounds at a single concentration of 5 µM revealed that four of them inhibited fungal growth for at least 60% (Figure 1) (Appendix A). The value of 60% of inhibition was chosen because posaconazole, which was used as the standard antifungal drug, inhibits 56% of *R. oryzae* growth (Table 1).

The four compounds were identified as alexidine (MMV396785), an antimicrobial agent of the class of bis-biguanide; MMV1580844, a non-commercial antibacterial drug; and MMV642550 and MMV019724, two non-commercial antiviral drugs (Table 1). The chemical structures of the compounds are shown in Figure 2. Fungal growth inhibition of 70% was observed for alexidine and MMV019724, whereas MMV1580844 inhibited 71% and MMV642550 inhibited 60% of fungal growth (Table 1). Fungal viability was reduced by about 89, 88, 59 and 77% in the presence of alexidine, MMV1580844, MMV642550 and MMV019724, respectively (Table 1).

### 3.2. Minimum Inhibitory and Fungicidal Concentrations of the Selected Compounds

Since only a single concentration of each compound was used in the screening step, minimum inhibitory (MIC) and fungicidal (MFC) concentrations of the selected compounds were determined for *R. oryzae*, *R. microsporus* and *R. stolonifer*. Alexidine MIC and MFC varied between 0.63–1.25 µM and 1.25–10 µM, respectively (Table 2). MIC values of the non-commercial drugs were observed at 0.08, 2.5–5.0 and 0.63–1.25 µM for MMV1580844, MMV642550 and MMV019724, respectively (Table 2). Regarding MFCs of the non-commercial drugs, all of them were found to be >20 µM for all three *Rhizopus* species tested (Table 2).

Posaconazole and amphotericin B were used as the standard antifungal drugs. Posaconazole displayed MIC values between 0.63–2.5 µM and MFC > 20 µM for all fungal species (Table 2). Amphotericin B was inefficient against these fungi, with both MIC and MFC being >20 µM (Table 2).

### 3.3. Effect of Selected Compounds on Rhizopus spp. Biofilms

For selected compounds, the effect on biofilm formation and mature biofilms was evaluated for *R. oryzae*, *R. microsporus* and *R. stolonifer*. Regarding biofilm formation, alexidine decreased *R. oryzae* and *R. microsporus* biomass, extracellular matrix and viability to 25% when MIC was used (Figure 3A–F). However, for *R. stolonifera*, similar results were only observed at 2× MIC or higher concentrations (Figure 3G–I). For the non-commercial drugs, a significant reduction in *R. oryzae* and *R. microsporus* biofilm parameters was more prominent, especially when 2× MIC or higher concentrations were used (Figure 3A–F), whereas 1× MIC was more effective in reducing *R. stolonifer* biofilm parameters (Figure 3G–I).

Regarding mature biofilms, it was evaluated whether the selected compounds could degrade biofilms formed prior to drug treatment. Alexidine led to a significant decrease in biofilm parameters at 2× MIC for *R. oryzae* and *R. microsporus* (Figure 4A–F) and at 4× MIC for *R. stolonifer* (Figure 4G–I). Non-commercial drugs reduced biofilm parameters at 2–8× MIC for *R. oryzae* (Figure 4A–C) and at 1–8× MIC for *R. microsporus* and *R. stolonifer* (Figure 4D–I).

### 3.4. Morphological Alterations Caused by Selected Compounds Evaluated by SEM

To elucidate how these compounds affect *Rhizopus* cells, SEM analysis was performed to evaluate fungal morphology. *R. oryzae* was chosen as the representative species, since it is the most frequent species associated with *Rhizopus* infections. Each compound was used at MIC_50_ (Table 2) and fungi were incubated for 48 h prior to SEM procedures. *R. oryzae* grown in the absence of any compound showed thicker hyphae and the formation of sporangia and spores, typical of Mucorales growth (Figure 5). When cells were treated with alexidine, abnormal sporangia were observed, presenting wrinkled spores that germinated before detaching from the sporangium (Figure 5). The treatment with MMV1580844 led to the formation of rare and empty sporangia (Figure 5). In the presence of MMV642550, no sporangia were observed and abnormal branches were seen along the mycelia (Figure 5). Regarding the treatment with MMV019724, deformed sporangia and spores were observed (Figure 5).

### 3.5. Influence of Selected Drugs in Cellular Parameters

To investigate the alterations on fungal cells caused by the selected drugs, we used fluorescent probes to measure the chitin content (calcofluor white), DNA (Sytox blue), oxidative stress (DCFDA), mitochondrial membrane potential (JC-1) and neutral lipids (Nile Red). *R. oryzae* was used as the representative species. Alexidine treatment led to a reduction in the amounts of chitin, DNA and neutral lipids (Figure 6A,B,E), suggesting modifications affecting both cell surface and nucleic acid. MMV1580844 induced oxidative stress and the depolarization of mitochondrial membranes (Figure 6C,D). MMV642550 decreased chitin and neutral lipid contents (Figure 6A,E). MMV019724 treatment resulted in lower levels of chitin, neutral lipids and mitochondrial alterations (Figure 6A,D,E). Together, these effects reflected that those compounds affected the fungal cell surface and their metabolism.

### 3.6. Interaction of Selected Compounds with Antifungal Drugs

Considering that the selected compounds display antifungal activity against *Rhizopus* species and that they cause alterations in fungal cells, we decided to investigate whether the compounds would interact with antifungal agents already used in clinical settings. Each compound selected was combined with either posaconazole or amphotericin B, two of the most frequently used antifungals to treat Mucorales infections. Using the FIC index method, which is based on the reduction in MIC values, none of the combinations resulted in a synergistic effect, because all MIC values of combined drugs remained the same as the drugs alone (Table 3).

On the other hand, when data were analyzed using the BLISS method, which considers the percentage of efficacy, amphotericin B acted synergistically with alexidine, MMV1580844 and especially with MMV642550 (Table 4), since the percentage of fungal inhibition caused by sub-inhibitory concentrations of each drug increased when drugs were employed in combination. However, posaconazole displayed antagonistic effects in combination with all selected compounds, especially with MMV1580844 and MMV642550 (Table 4). More studies are needed to clarify how these compounds interact with antifungal drugs.

### 3.7. Cytotoxicity and Selectivity Index of Selected Compounds

The cytotoxicity of alexidine, MMV396785, MMV1580844, MMV642550 and MMV019724 was evaluated using three cell lineages: RAW, A549 and HaCaT. All drugs were found to be non-toxic for any cell line, because CC50 was found to be higher than 50 µM, the highest concentration tested (Table 5).

Considering the MIC values of each compound for *R. oryzae*, *R. microsporus* and *R. stolonifer*, the selectivity index (SI) was calculated. Alexidine showed an SI of 40.0 for planktonic cells of *R. oryzae* and *R. microsporus* and of 79.4 for *R. stolonifer*, whereas the SI in biofilms was found to be 20.0 for all fungal species (Table 5). MMV1580844 displayed an SI of 625 for planktonic cells of the three fungi, and for biofilms it varied between 156.3 and 312.5, except for *R. oryzae* whose biofilm was not inhibited by this drug (Table 5). MMV642550 presented an SI of 10–20 and 5–10 for planktonic cells and biofilms, respectively, except for biofilms of *R. oryzae* that were also not inhibited by this drug (Table 5). The selectivity index of MMV019724 was found to be 40.0–79.4 and 10–40 for planktonic cells and biofilms, respectively (Table 5). These results suggest that all drugs tested possess promising selectivity for fungal cells.

### 3.8. In Silico Analysis of Drug-Likeness

One of the most important tools in the early stages of drug discovery is to predict whether a compound is orally well absorbed. Considering that our screening revealed three novel chemical compounds with relevant antifungal activity, we decided to determine their drug-likeness based on the main guidelines for predicting oral absorption and bioavailability employing Lipinski’s Rule of Five (RoF) [27] and Veber’s rule [28]. Although this approach is suitable for studying novel molecules, alexidine was also included in our analysis due to the lack of data related to its pharmacokinetic properties. In addition, posaconazole and amphotericin B were added for comparative purposes.

As shown in Table 6, MMV1580844 and MMV642550 completely satisfied the Ro5 (MW ≤ 500 Da, cLogP ≤ 5, HBD ≤ 5 and HBA ≤ 10) and MMV019724 violated one of the parameters (cLogP > 5), whereas alexidine displayed two violations (MW > 500 and HBD > 5). As specified by the guideline, compounds that exhibit two or more violations of any of the four parameters are more likely to have low permeability or poor absorption, leading to poor bioavailability. Additionally, only alexidine exceeded the threshold for good oral bioavailability established by Veber’s rule. The results suggest that the three novel chemical compounds MMV1580844, MMV642550 and MMV019724 possess desirable drug-likeness properties of an oral drug candidate. Interestingly, these compounds showed better compliance with RoF and Veber’s rule than posaconazole and amphotericin B, the antifungal agents commonly used to treat Mucorales infections.

## 4. Discussion

The screening of compound collections has been an important tool to identify new drugs for different fungal infections [12,13,14,29], especially mucormycosis that is a hard-to-treat disease. The present study screened the Pandemic Response Box^®^ collection from MMV aiming to identify drugs that are active against the *Rhizopus* species. *Rhizopus oryzae* was chosen as the reference strain since it is responsible for the majority of mucormycosis cases worldwide [2,30,31,32]. We identified four compounds that inhibited at least 60% of fungal growth at a concentration of 5 µM. The selected compounds were identified as alexidine (MMV396785), an antimicrobial agent of the class of bis-biguanide; a non-commercial antibacterial drug (MMV1580844); and two non-commercial antiviral drugs (MMV642550 and MMV019724).

Alexidine is not a conventional antifungal drug, but it is used as an antimicrobial agent in solutions for contact lenses and mouthwashes. It is a cationic compound that binds to the negatively charged bacterial cell wall molecules such as lipopolysaccharide and lipoteichoic acid [33]. MMV1580844 is an antibacterial diaminopyridine propargyl-linked antifolate, which targets dihydrofolate reductase (DHFR) in mammalian and yeast cells [34,35,36]. DHFR is a potent target for antimicrobial therapy, since it effectively blocks thymidine synthesis, thus leading to cell death. DHFR inhibitors are effective tools for both prokaryotic and protozoal pathogens but are not used in the treatment of invasive fungal infections, since DHFR is also essential in human cells. Therefore, propargyl-linked antifolate compounds must be designed with selective inhibition of the pathogenic enzyme [34,35,37,38]. MMV642550 is an 8-hydroxyquinoline derivative that functions as a platform for several FDA-approved drugs and is a known inhibitor for viral replication [39]. MMV019724 is also an 8-hydroxyquinoline derivative, showing antiviral, antifungal and anti-amoebic effects via lactate dehydrogenase inhibition [13,20,29,38,40].

All selected compounds were able to inhibit the growth of the three *Rhizopus* species tested. Although the non-commercial drug MMV1580844 showed the lowest MIC value, only alexidine was able to kill *R. oryzae* and *R. stolonifer* at 10 µM, and *R. microsporus* at 1.25 µM. Previous studies on alexidine reported that this drug has broad-spectrum antifungal activity, inhibiting the growth of several *Candida* species (including *C. albicans* and *C. auris*), *Cryptococcus neoformans*, *Aspergillus fumigatus* and several species from the Mucorales order such as *R. oryzae*, *Mucor circinelloides* and *L. corymbifera* [11]. Furthermore, alexidine showed activity against other species of filamentous fungi such as *A. calidoustus*, *Fusarium solani* and *F. oxysporum*, as well as *L. prolificans* and *S. apiospermum* [11,41]. The non-commercial drug MMV1580844 was also active against bacteria and protozoa species, such as *Staphylococcus aureus* and *Balamuthia mandrillaris*, and some *Candida* species, such as *C. albicans*, *C. glabrata* and *C. auris*, with the MIC value being lower than 100 nM for all of them [35,36,40,42]. The quinoline analogue MMV019724 showed potent in vitro activity against clinically important yeasts, such as *C. auris* and *Cryptococcus* species, being able to inhibit 90% of growth at 2.5 and 5 µM, but it did not kill these yeasts [13], as observed for the *Rhizopus* species in our results. In the *Sporothrix* species, MMV019724 inhibits 50% of *S. brasiliensis* and *S. schenckii* at 0.25 and 1 µM, respectively [20]. Despite being the drug of choice for mucormycosis treatment, amphotericin B (used as a control) was unable to inhibit the growth of the three *Rhizopus* species tested in this work. It has already been shown that although amphotericin B is one of the drugs of choice for mucormycosis, its activity varies depending on the genus, species or isolate, which helps to explain why the strains used in the present study were shown to be resistant [43]. On the other hand, posaconazole showed low MIC values, but did not show fungicidal activity on these *Rhizopus* species. 

Species from the Mucorales order, such as *R. oryzae*, *L. corymbifera* and *R. pusillus*, produce robust biofilm with filamentous and adherent structures enclosed by an extracellular matrix composed primarily of glucosamine (GlcN) and *N*-acetylglucosamine (GlcNAc) [44]. Since Mucorales biofilm formation may be involved in the pathogenesis of paranasal fungal balls, endocarditis, osteomyelitis and catheter-based infections [44], the present study analyzed the inhibitory effects of the four selected compounds from the Pandemic Response Box^®^, both for their abilities to inhibit *R. oryzae*, *R. microsporus* and *R. stolonifer* biofilm formation and for their efficacies against preformed biofilms. Alexidine and the three non-commercial drugs inhibited the biofilm formation, with alexidine being the most potent compound inhibiting almost 100% of the total biomass, matrix and viability, and the other three compounds inhibiting more than 50% of biofilm formation. As expected, mature biofilms were more resistant to all compounds when compared to biofilm formation. Mamouei and colleagues also reported the anti-biofilm potential of alexidine, since it inhibited the biofilm formation of *A. fumigatus*, *C. neoformans* and *Candida* species. Alexidine was also effective against preformed biofilms from *C. albicans* in vitro and in vivo using a catheter model [11]. To our knowledge, the present study showed for the first time the ability of MMV1580844, MMV642550 and MMV019724 to inhibit fungal biofilms.

Scanning electron microscopy (SEM) of treated *R. oryzae* cells was performed to evaluate morphological alterations caused by the treatment with the four selected compounds. Our results showed that all tested compounds mainly affected sporangia and spore formation. In addition, fluorescent probes were used to analyze the effects of the selected compounds on *R. oryzae* cells. Alexidine treatment reduced the chitin and neutral lipid content, suggesting modifications on the fungal cell surface, as well as a reduction in the DNA content. Although alexidine displays antifungal activity, its mechanism of action in fungi is still unclear. Being negatively charged, alexidine could be attracted to the negatively charged fungal cell wall, thus inducing lipid phase separation and lipid raft formation, as shown for bacterial membranes [33]. Previous reports demonstrated that alexidine causes mitochondrial damage by targeting mitochondrial phosphatase PTPMT1 in cancer cell lines [45,46]. However, in our study, alexidine did not affect the mitochondrial membrane potential of *R. oryzae*.

All three non-commercial compounds tested in this work did not affect DNA content. MMV1580844 increased ROS production and decreased the mitochondrial membrane potential. Antifolate drugs, similar to MMV1580844, can tightly bind to DHFR and inhibit DNA synthesis and cell proliferation [47]. Previous work showed that antifolate drugs can also be used as potent antitumor drugs and induce oxidative DNA damage [47]. MMV642550 and MMV019724 are quinoline analogues and decreased chitin and neutral lipid content, but only MMV019724 affected the mitochondrial membrane potential of *R. oryzae*.

Synergistic effects among the selected compounds and the two main antifungals used for Mucorales infections were analyzed using the FIC index method and the BLISS method. According to the FIC index method, which considers the MIC value reduction, none of the combinations were synergic. However, the BLISS method considers the growth inhibition efficacy of the combined drugs, and the combination between alexidine, MMV1580844 and MMV642550 with amphotericin B was synergic. The combination between MMV019724 and amphotericin B showed antagonistic effects. All combinations of the compounds and posaconazole were antagonistic. Mamouei and colleagues demonstrated that alexidine enhanced the effect of fluconazole against *C. albicans* [11]. However, little is known about the interaction potential of the four selected compounds with other drugs used in clinical settings. Therefore, more studies are needed to clarify the consequences of combining these molecules with other therapeutic agents.

Importantly, cytotoxicity results reported in this work demonstrated that alexidine, MMV1580844, MMV642550 and MMV019724 were non-toxic for the three mammalian cells tested, because the CC_50_ was higher than 50 µM, which is considerably higher than MIC_50_ values for planktonic cells and the preformed biofilm of the three *Rhizopus* species tested. MMV1580844 showed a higher selectivity index (SI) for planktonic cells as well as preformed biofilm, and was considered more selective for fungi than mammalian cells. Rice and colleagues also observed the non-toxic effect of MMV1580844 on the A549 cell line, with IC_50_ being higher than 10 µM [40]. In contrast, the authors observed lower IC_50_ (6.2 µM) for MMV019724 when tested with the A549 cell line [40]. Previous work reported that alexidine damaged HUVEC, an endothelial cell line, and A549 cells at 14.7 µg/mL (28.89 µM) [11]. These results for A549 are different from ours, probably due to differences in the methodologies used to measure cytotoxicity. 

During the early stages of drug development, the estimation of pharmacokinetic properties has become a crucial tool for improving the success rate of drug discovery programs [48]. One of the most important pharmacokinetic properties of a compound is high oral bioavailability, considering the advantages of oral drug administration (such as convenience, non-invasiveness, high patient compliance and cost-effectiveness) [49]. A common and simple approach to predict oral bioavailability and absorption of a compound is to analyze its physicochemical properties using cheminformatics tools [21]. In the present work, we focused on estimating the drug-likeness of the selected compounds based on a set of physicochemical properties, aiming at a more effective prospection of these molecules as novel potential candidates for oral antifungal agents. Moreover, alexidine, a well-known bis-biguanide, was included in our in silico analysis given the lack of data related to its pharmacokinetic profile. 

The physicochemical properties analyzed include some molecular descriptors present in the guideline proposed by Pfizer researchers Lipinski and colleagues and known as the “Rule of Five” (RoF) [27]. According to the RoF, a compound is more likely to show good absorption and permeability if it either completely satisfies or only violates one of the following parameters: MW ≤ 500 Da, cLogP ≤ 5, HBD ≤ 5 and HBA ≤ 10. Our analysis revealed that MMV1580844 and MMV642550 fully complied with Ro5, whereas MMV019724 violated only one parameter (cLogP > 5). These compounds showed drug-like physicochemical properties common with approximately 90% of commercial oral drugs [27,50], and good oral bioavailability may be expected. On the other hand, alexidine violated two parameters (MV > 500 Da and HBD > 5), suggesting that it may have low permeability and, consequently, may not be well absorbed. In fact, many HBDs in the chemical structure of the compound have a deleterious effect on its solubility, permeability and bioavailability [51,52,53]. Notably, only 5% of oral drugs have HBD > 5 [54], highlighting the importance of this molecular descriptor for predicting drug absorption. Additionally, although MW plays an important role in determining the permeability of drugs across the intestinal membrane [55], a small number of approved oral drugs are beyond the chemical space defined by Ro5 (bRoF) and possess a high MW, such as isavuconazonium, an oral prodrug of isavuconazole [50,56,57].

Furthermore, other descriptors were also analyzed, such as the number of rotatable bonds (nRotBs) and the topological polar surface area (TPSA). According to GlaxoSmithKline researchers Veber and colleagues, both descriptors are important predictors of good oral bioavailability, since highly flexible (nRotB > 10) and highly polar compounds (TPSA > 140 Å^2^) are considered to be less membrane permeable [28,58]. Among the four selected compounds, only alexidine exceeded the limit established by Veber’s rule for oral bioavailability, whereas MMV1580844, MMV642550 and MMV019724 met all criteria established by this guideline. Interestingly, these three novel compounds showed better drug-likeness properties than those observed for the standard drugs posaconazole and amphotericin B, which did not fully comply with Ro5 (two and three violations, respectively) or Veber’s rule. Indeed, absolute bioavailability of posaconazole after oral administration has been estimated to range from 8% to 47% [59], whereas amphotericin B has negligible oral bioavailability [60].

Therefore, based on the set of molecular descriptors analyzed, MMV1580844, MMV642550 and MMV019724 have desirable drug-likeness properties of drug candidates and would be easily orally absorbed, whereas alexidine violates conventional guidelines for drug-likeness. Nevertheless, our results must be further examined through in vitro and in vivo approaches to investigate the potential of these compounds as future options for the oral treatment of mucormycosis.

The present study identified four promising compounds from the Pandemic Box^®^ library, which displayed interesting cellular changes in *Rhizopus* spp., as well as good in silico parameters. These results contribute to the acknowledgement of new candidates to improve the treatment of mucormycosis in the future.

## Figures and Tables

**Figure 1 jof-09-00187-f001:**
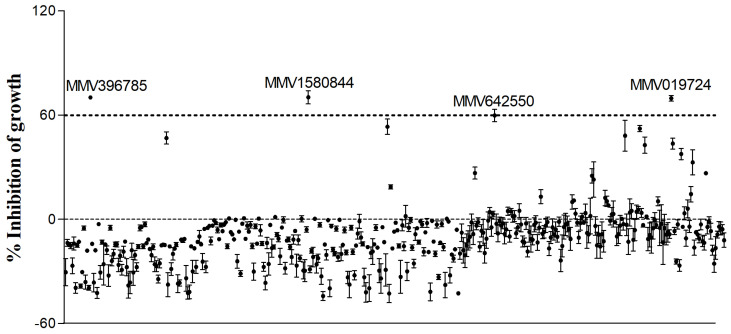
Screening of the Pandemic Response Box^®^ library. A total of 400 compounds were tested against *R. oryzae* UCP1295. Fungal growth was quantified after incubation for 48 h via optical density and those presenting at least 60% of inhibition (dotted line) were selected.

**Figure 2 jof-09-00187-f002:**
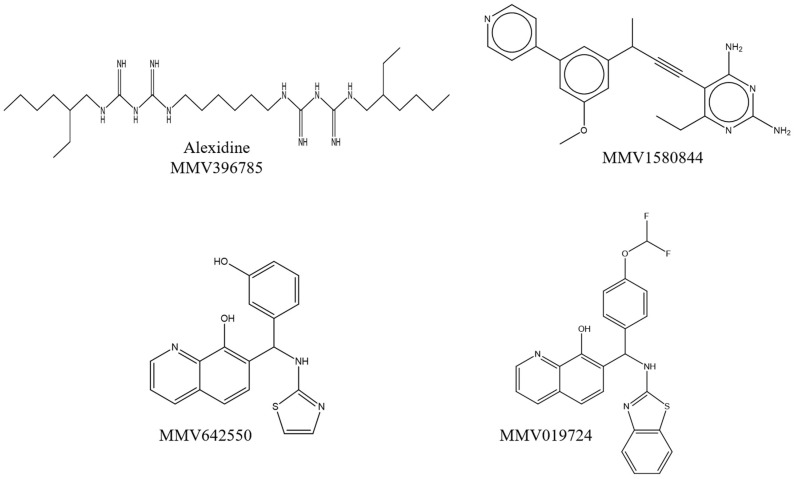
Molecular structures of the effective compounds against *R. oryzae* UCP1295. The compounds present in the Pandemic Response Box^®^ library are MMV396785 (Alexidine), MMV1580844, MMV642550 and MMV019724.

**Figure 3 jof-09-00187-f003:**
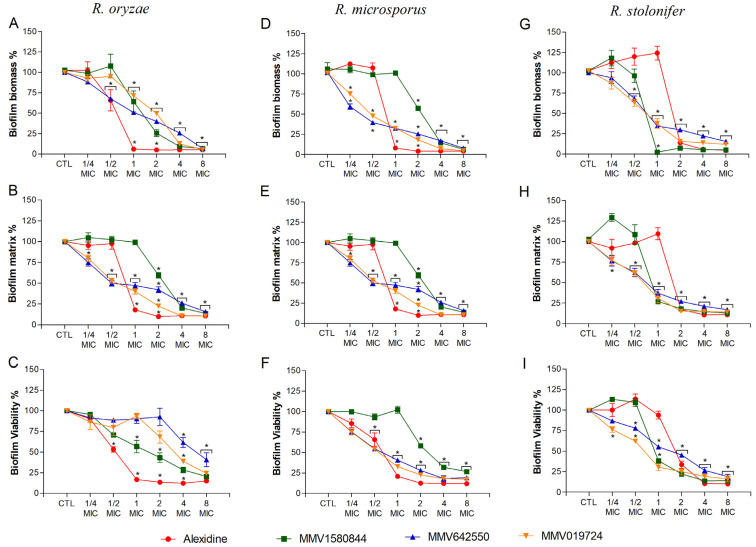
Effect of alexidine, MMV1580844, MMV642550 and MMV019724 on biofilm formation of *Rhizopus* species. Fungal cells were grown on polystyrene surface in the presence of different concentrations of selected compounds (¼–8× MIC). Fungal biomass (**A**,**D**,**G**), extracellular matrix (**B**,**E**,**H**) and viability (**C**,**F**,**I**) were measured using violet crystal, safranin and XTT reduction assay, respectively. * *p* < 0.05, compared to zero (absence of drug) for each species.

**Figure 4 jof-09-00187-f004:**
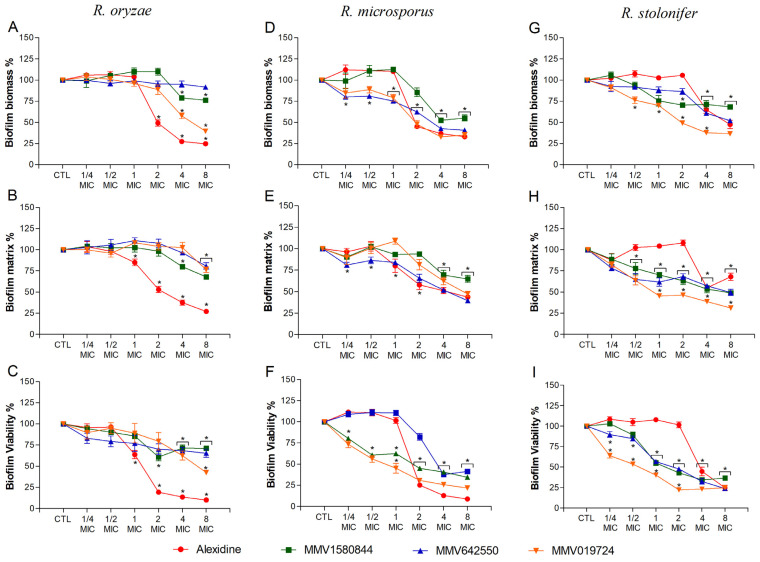
Effect of alexidine, MMV1580844, MMV642550 and MMV019724 on preformed biofilms of three *Rhizopus* species. Fungal biofilm was firstly formed in RPMI 1640 medium on polystyrene surface for 24 h and then it was treated with different concentrations of selected compounds (¼–8× MIC) for another 24 h incubation. Intact fungal biofilms were considered controls (CTL, 100% biofilm) and their degradation due to treatment was compared to the control. Fungal biomass (**A**,**D**,**G**), extracellular matrix (**B**,**E**,**H**) and viability (**C**,**F**,**I**) were measured using violet crystal, safranin and XTT reduction assay, respectively. * *p* < 0.05, compared to zero (absence of drug) for each species.

**Figure 5 jof-09-00187-f005:**
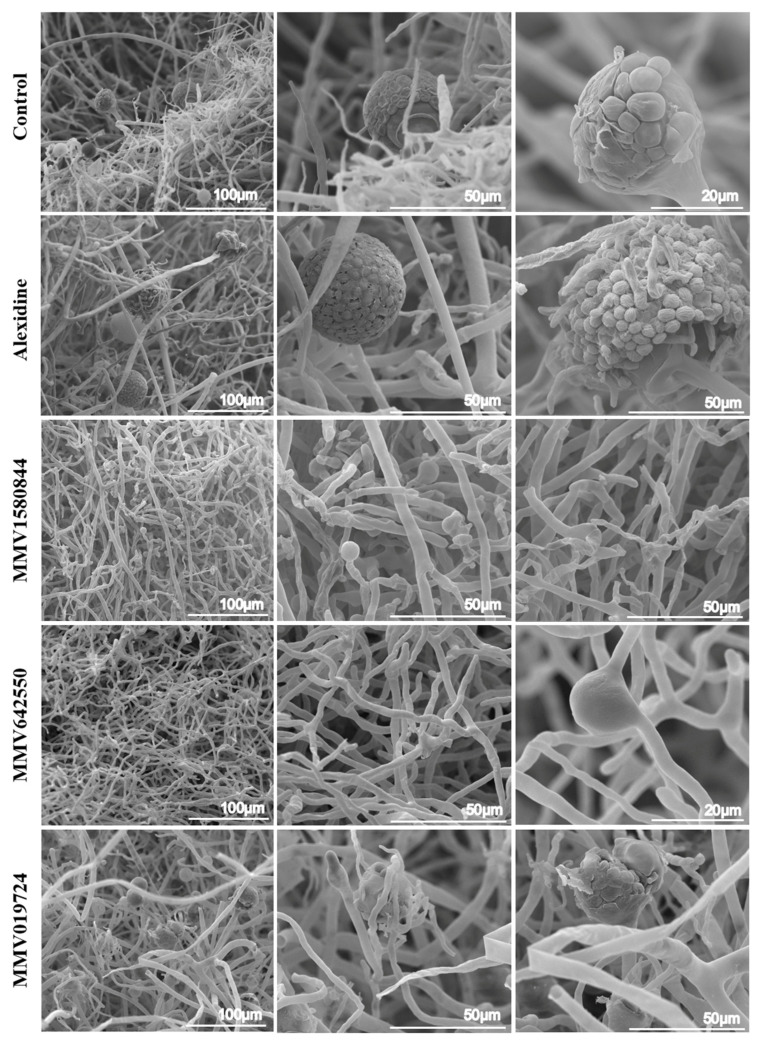
Ultrastructural alterations of *R. oryzae* UCP1295 after exposure to alexidine, MMV1580844, MMV642550 and MMV019724, evaluated via scanning electron microscopy. Untreated cells (control) exhibit thicker hyphae and the formation of sporangia and spores, typical of Mucorales growth, whereas samples treated with MIC_50_ (Table 2) values of each selected compound for 48 h show alterations in fungal surface.

**Figure 6 jof-09-00187-f006:**
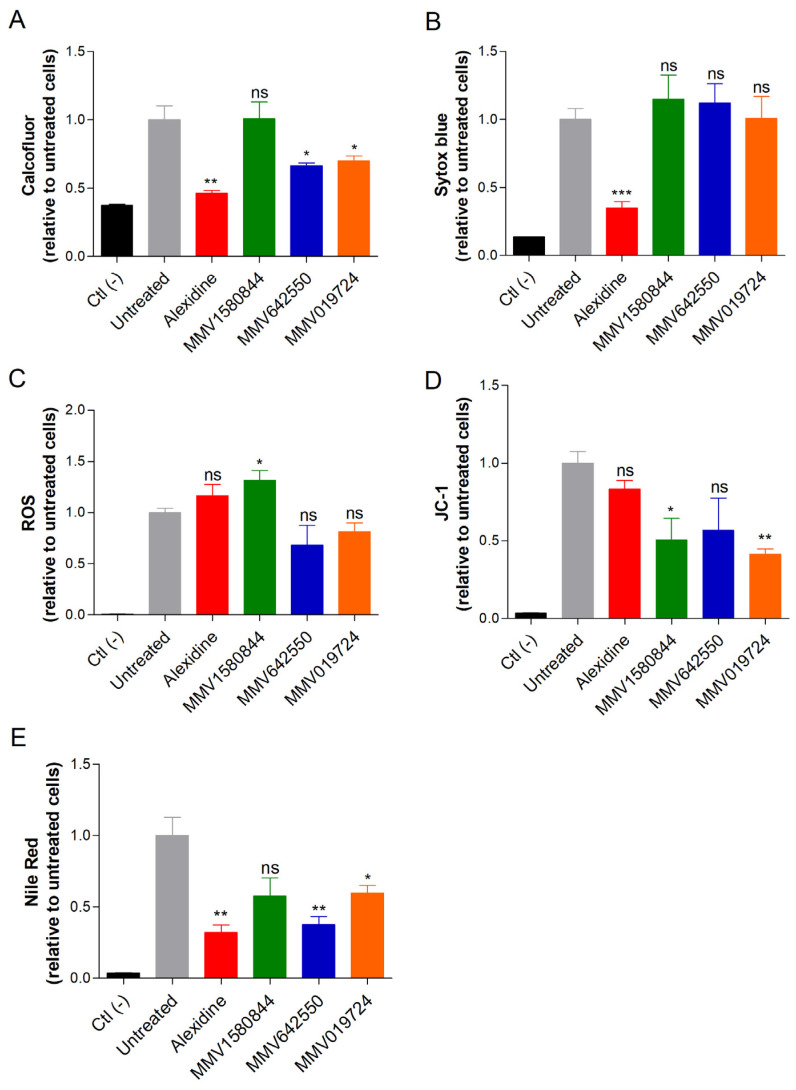
The effect of alexidine, MMV1580844, MMV642550 and MMV019724 on *R. oryzae* cells analyzed using fluorescent staining. Cells were grown in the presence of ½× MIC_50_ of each selected compound for 48 h at 37 °C. Chitin content was analyzed using calcofluor white (**A**). Intracellular DNA quantification was analyzed using Sytox blue staining (**B**). Oxidative stress; ROS was measured using DCFH-DA (**C**). The mitochondrial membrane polarization was measured using JC-1 (**D**). Neutral lipids were quantified using Nile Red stain (**E**). Ctl (-), a negative control that represents cells in the absence of fluorescent stain. Untreated, a positive control that represents cells stained with fluorescent stain, but without drug treatment. * *p* < 0.05; ** *p* < 0.01; *** *p* <0.001; ns—not significant.

**Table 1 jof-09-00187-t001:** Effective selected compound from Pandemic Response Box^®^ at 5 µM against *R. oryzae* UCP1295.

Compound Code	Growth Inhibition	Viability Inhibition (XTT)	Name or ID ChEMBL	Disease Area
MMV396785	70%	89%	Alexidine	Biguanide antimicrobial
MMV1580844	71%	88%	CHEMBL2335419	Antibacterials
MMV642550	60%	59%	CHEMBL1426340	Antiviral
MMV019724	70%	77%	CHEMBL548113	Antiviral
--	56%	73%	Posaconazole	Azole Antifungals

**Table 2 jof-09-00187-t002:** Minimum inhibitory and fungicidal concentration of alexidine, MMV1580844, MMV642550, MMV019724, posaconazole and amphotericin B against three *Rhizopus* species.

Compound or Code	*R. oryzae*		*R. microsporus*		*R. stolonifer*	
	MIC_50_	MFC	MIC_50_	MFC	MIC_50_	MFC
Alexidine	1.25 µM	10 µM	1.25 µM	1.25 µM	0.63 µM	10 µM
MMV1580844	0.08 µM	>20 µM	0.08 µM	>20 µM	0.08 µM	>20 µM
MMV642550	2.5 µM	>20 µM	5 µM	>20 µM	2.5 µM	>20 µM
MMV019724	0.63 µM	>20 µM	1.25 µM	>20 µM	1.25 µM	>20 µM
Posaconazole	1.25 µM	>20 µM	2.5 µM	>20 µM	0.63 µM	>20 µM
Amphotericin B	>20 µM	>20 µM	>20 µM	>20 µM	>20 µM	>20 µM

MIC: minimal inhibitory concentration; MFC: minimal inhibitory fungicidal concentration.

**Table 3 jof-09-00187-t003:** Antifungal activity of alexidine, MMV1580844, MMV642550, MMV019724, posaconazole and amphotericin B—alone and in combinations according to the Fractional Inhibitory Concentration Index (FICI)—against *R. oryzae* UCP1295. MIC values were used to analyze the interaction between alexidine, MMV1580844, MMV642550 and MMV019724 with posaconazole or amphotericin B.

MIC_50_ Alone (µM)		MIC_50_ Combined (µM)		FICI
Alexidine	1.25	Alexidine/Posa	1.25/5.0	2.0 (no effect)
MMV1580844	0.08	MMV1580844/Posa	0.08/1.25	2.0 (no effect)
MMV642550	2.5	MMV642550/Posa	2.5/5.0	2.0 (no effect)
MMV019724	0.63	MMV019724/Posa	0.63/1.25	2.0 (no effect)
Posa	1.25	Alexidine /AmphoB	1.25/40	2.0 (no effect)
AmphoB	40	MMV1580844/AmphoB	0.08/40	2.0 (no effect)
--	--	MMV642550/AmphoB	2.5/40	2.0 (no effect)
--	--	MMV019724/AmphoB	0.63/40	2.0 (no effect)

MIC: minimal inhibitory concentration; Posa: posaconazole; AmphoB: amphotericin B.

**Table 4 jof-09-00187-t004:** Antifungal activity of alexidine, MMV1580844, MMV642550 and MMV019724 in combinations with amphotericin B and posaconazole according to the Bliss independence model.

			Efficacy of Combined Drugs					
	Efficacy of Drugs Alone (% of Inhibition)		Amphotericin B			Posaconazole		
	MIC_50_	½ MIC_50_	*E* _obs_	*E* _exp_	Δ*E*, % (Interaction)	*E* _obs_	*E* _exp_	Δ*E*, % (Interaction)
Alexidine	67.6	21.5	85.9	84.2	1.7 (**S**)	91.9	96.6	−4.7 (**A**)
MMV1580844	63.5	12.6	60.5	58.2	2.3 (**S**)	60.2	77.6	−17.4 (**A**)
MMV642550	62.9	42.3	78.1	54.6	23.5 (**S**)	60.0	86.5	−26.5 (**A**)
MMV019724	62.0	32.5	74.8	82.1	−7.3 (**A**)	58.0	63.6	−5.6 (**A**)
AmphoB	79.6	21.4	NP	NP	NP	NP	NP	NP
Posa	63.4	6.5	NP	NP	NP	NP	NP	NP

MIC, minimal inhibitory concentration. *E*_obs_, efficacy observed in the analysis. *E*_exp_, efficacy expected according to Bliss calculation. Δ*E*, difference between *E*_obs_ and *E*_exp_. NP, not performed. **S**, synergistic interaction. **A**, antagonist interaction.

**Table 5 jof-09-00187-t005:** Cytotoxicity values and selectivity index (SI) of selected compounds with activity against planktonic cells and preformed biofilm of *Rhizopus* species.

		Selectivity Index (SI)					
Compound (µM)	CC_50_ *^a^*	*R. oryzae*		*R. microsporus*		*R. stolonifer*	
		Planktonic Cell *^b^*	Preformed Biofilm *^c^*	Planktonic Cell *^b^*	Preformed Biofilm *^c^*	Planktonic Cell *^b^*	Preformed Biofilm *^c^*
Alexidina MMV396785	>50	>40	>20	>40	>20	>79.4	>20
MMV1580844	>50	>625	ND	>625	>156.3	>625	>312.5
MMV642550	>50	>20	ND	>10	>5	>20	>10
MMV019724	>50	>79.4	>10	>40	>40	>40	>40

ND: Not detected. *^a^* CC_50_ for RAW, A549 and HaCaT cell lines was the same. *^b^* MIC_50_ of the planktonic cells. *^c^* The SI for preformed biofilm was calculated based on biofilm viability parameter.

**Table 6 jof-09-00187-t006:** Physicochemical properties of the selected compounds after screening to predict their drug-likeness (based on Lipinski’s Rule of Five and Veber’s rule).

Compound	Lipinski’s RoF					Veber’s Rule	
	MW	cLogP	HBA	HBD	nViol	TPSA (Å^2^)	nRotB
Alexidine	581.71	4.88	4	6	2	177.6	23
MMV1580844	373.45	3.76	4	2	0	99.94	4
MMV642550	349.41	3.69	4	3	0	106.5	4
MMV019724	449.47	5.38	6	2	1	95.51	6
Posaconazole	700.78	4.23	12	1	2	111.7	12
Amphotericin B	924.08	0.32	18	12	3	319.6	3

RoF—rule of five; MW—molecular weight; cLogP—calculated octanol/water partition coefficient; HBA—number of hydrogen bond acceptors; HBD—number of hydrogen bond donors; nViol—number of Lipinski’s RoF violations; TPSA—topological polar surface area; nRotB—number of rotatable bonds.

## Data Availability

Not applicable.

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
