# Peer review of "Promising Antifungal Molecules against Mucormycosis Agents Identified from Pandemic Response Box®: In Vitro and In Silico Analyses"

_jof, 2023, doi:10.3390/jof9020187_

Round 1

Reviewer 1 Report

The manuscript entitled “Promising antifungal molecules against mucormycosis agents identified from Pandemic Response Box® – in vitro and in silico analyses” is about to provide Pandemic Response Box® as a new antifungal molecules.

This manuscript was written well and experiments were done properly, but need some modification to be published.

At the “Antifungal susceptibility testing”, It is not clear because authors said that MIC is actually MIC50 not MIC100. Then authors use all tested sample to see the MFC? then it is hard to compare the meaning of MFC. Authors also need provide the MIC100 then compare with MFC. And, bacteria should be mistake.

Authors need to provide the source of material (company, city and nations) used in this study for all materials for reproducibility.

At line 147, CO2 should be CO2.

At line 154, R. oryzae should be italicized.

At line 155, RPMI is just RPMI or Supplemented RPMI?

Authors need enough references for material methods. Some do not have reference and some need more because authors provide only 1 reference but said “modified”.

Reference style is not consistence in line 166.

At line 260, need “and” between 59 and 77%.

At line    HaCat should be HaCaT.

In the case of “mature biofilms”, is it to check the degradation of biofilm formed or to check the reiniation of biofilm formation? it is not clear and also the meaning of this experiments. based on that, authors need the difference of initial biomass and after 24h at the figure 4.

At the figure 4, please double check the E and H.

What is the meaning of MIC50 at line 398 and 433? Authors already use fold, but why use like that? and also there is other meaning.

It is curious how authors calculate the SI value because Cell toxicity is higher than 50 that obtained by experiments.

Authors need to explain what make the differences between the screening results and MIC values.

Authors need to explain why amphotericin B did not work in this manuscript, because authors use that as a positive control, but not work.

How authors think that the inhibition of growth reduce the biofilm formation or there is other reason?

Species name must be italicized in reference.

Author Response

Dear Reviewer,

We appreciated all comments made in our manuscript. It will certainly improve its quality. Please find below a point-by-point answer to your suggestions.

1) At the “Antifungal susceptibility testing”, It is not clear because authors said that MIC is actually MIC50 not MIC100. Then authors use all tested sample to see the MFC? then it is hard to compare the meaning of MFC. Authors also need provide the MIC100 then compare with MFC. And, bacteria should be mistake.

ANSWER: During the experiments, none of the concentrations reached 100% of fungal inhibition, even the standard drugs (posaconazole and amphotericin B). It might happen because these fungi are known to be very resistant to all antifungal drugs. Considering the resistance pattern of them, we decided to use MIC50, which represent the inhibition of half of fungal growth and is relevant for the study. It was clarified in the text (lines 124-125). For the MFC, all concentrations of the serial dilution were used to evaluate the fungal growth on solid medium. This information was corrected in the text (lines 127-128). The expression “bacterial growth” was a mistake and it was modified in the text (line 129).

2) Authors need to provide the source of material (company, city and nations) used in this study for all materials for reproducibility.

ANSWER: We checked the source of the materials used in the study and added those that were missing.

3) At line 147, CO2 should be CO2.

ANSWER: It was corrected in the text (line 154).

4) At line 154, R. oryzae should be italicized.

ANSWER: It was corrected in the text (line 162).

5) At line 155, RPMI is just RPMI or Supplemented RPMI?

ANSWER: Supplemented RPMI was used for all experiments to avoid changes in the results. The information was corrected in the text (line 163).

6) Authors need enough references for material methods. Some do not have reference and some need more because authors provide only 1 reference but said “modified”.

ANSWER: References in the Material and Methods section were checked and some new citations were added to provide more information.

7) Reference style is not consistence in line 166.

ANSWER: Reference style was corrected (lines 173-174).

8) At line 260, need “and” between 59 and 77%.

ANSWER: It was corrected in the text (line 270).

9) At line    HaCat should be HaCaT.

ANSWER: It was corrected in the text (line 201).

10) In the case of “mature biofilms”, is it to check the degradation of biofilm formed or to check the reiniation of biofilm formation? it is not clear and also the meaning of this experiments. based on that, authors need the difference of initial biomass and after 24h at the figure 4.

ANSWER: The effect of the compounds on mature biofilms was evaluate to check the degradation of biofilms formed prior to the treatment. The control (CTL) was considered the intact biofilm and it was used to compare with biofilms in the presence of different concentrations of each compound. This information was clarified in the text (lines 321-322 and Figure 4 legend).

11) At the figure 4, please double check the E and H.

ANSWER: Sorry for this mistake. The Figure 4 was corrected.

12) What is the meaning of MIC50 at line 398 and 433? Authors already use fold, but why use like that? and also there is other meaning.

ANSWER: For these experiments, we used the concentrations that represent the MIC50 for each compound. Considering that each compound presented different MIC50 value, we standardized the representation to “MIC50” to avoid misunderstandings. MIC50 values can be found in Table 2, so it was added in the text (lines 402, 433 and 468).

13) It is curious how authors calculate the SI value because Cell toxicity is higher than 50 that obtained by experiments.

ANSWER: In fact, considering that the cell toxicity was found to be higher than 50, all SI values should be shown with the signal > to display a proper calculation. It was corrected in Table 5.

14) Authors need to explain what make the differences between the screening results and MIC values.

ANSWER: The screening was performed using a single concentration (5 µM) of each compound to check those presenting activity against R. oryzae (used as a reference species). Once we selected the compounds, they were tested in serially diluted concentrations to evaluate if lower concentrations could also be active. Thus, we found the MIC for the selected compounds and also analyzed the MFC. We added some information in the text to clarify it (lines 237 and 288).

15) Authors need to explain why amphotericin B did not work in this manuscript, because authors use that as a positive control, but not work.

ANSWER: Actually, amphotericin B was used as a standard, because it is one of the drugs of choice for the treatment of mucormycosis. However, some reports have been demonstrating that its activity depends on the genus, species or isolate, what could explain why our strains were resistant to amphotericin B. We added some explanations in the Discussion (lines 659-662).

16) How authors think that the inhibition of growth reduce the biofilm formation or there is other reason?

ANSWER: The formation of biofilms requires the cell growth, so it is expected that the compounds are able to reduce biofilm formation in concentrations that also inhibit fungal growth. In addition, alterations seen in cell parameters, especially those on fungal surface might also contribute to impair or difficult fungal adhesion, which is a crucial step to stimulate biofilm production. However, our data are not sufficient to elucidate it and more studies are needed to clarify how biofilm impairment occur when cells are treated with these compounds.

17) Species name must be italicized in reference.

ANSWER: It was corrected along the reference list.

Reviewer 2 Report

The work of da Silva Xisto and cols. is an interesting and robust original article in which the authors evaluated a library with 400 compounds called “Pandemic Response Box®” searching for possible candidates for mucormycosis. They identified four compounds with promising antifungal activity against Rhizopus spp. In general, the experimental design of the work, as well as the methods adopted was adequate and consistent with the planted objectives. Moreover, the manuscript is adequately structured and well written.

 The work is very informative and I recommend the acceptance in its present form.

Author Response

Dear Reviewer,

Thank you for your time spent reviewing our manuscript. Your evaluation and opinion is very important for the publication of our results. We believe these data will contribute to the field and for the development of further studies.

Best regards,

Authors.
